# Artificial Intelligence and Machine Learning in Ocular Oncology, Retinoblastoma (ArMOR): Experience with a Multiracial Cohort

**DOI:** 10.3390/cancers16203516

**Published:** 2024-10-17

**Authors:** Vijitha S. Vempuluru, Rajiv Viriyala, Virinchi Ayyagari, Komal Bakal, Patanjali Bhamidipati, Krishna Kishore Dhara, Sandor R. Ferenczy, Carol L. Shields, Swathi Kaliki

**Affiliations:** 1The Operation Eyesight Universal Institute for Eye Cancer, LV Prasad Eye Institute, Hyderabad 500034, India; vijitha.vempuluru@lvpei.org (V.S.V.); f20150131h@alumni.bits-pilani.ac.in (R.V.); 124003364@sastra.ac.in (V.A.); komal.bakal@ssei.ind.in (K.B.); 2The International Institute of Information Technology, Hyderabad 500032, India; patanjali.b@research.iiit.ac.in; 3The Center for Innovation, Bourntec, Madhapur, Hyderabad 500081, India; kishore.d@bourntec.com; 4Ocular Oncology Service, Wills Eye Hospital, Thomas Jefferson University, 840 Walnut Street, 14th Floor, Philadelphia, PA 19107, USA; sandor@shields.md (S.R.F.); carolshields@gmail.com (C.L.S.)

**Keywords:** eye, tumor, retinoblastoma, artificial intelligence, machine learning

## Abstract

**Simple Summary:**

In the last decade, artificial intelligence and machine learning (AI/ML) have been increasingly explored in the field of intraocular tumors. Retinoblastoma (RB) is the most common eye cancer in childhood. Early detection is crucial for optimizing outcomes in RB. Hence, we aimed to employ AI/ML to develop a potential screening tool for RB and established the feasibility of training an AI model to detect and classify RB from fundus images in an Asian Indian cohort previously. Taking this work ahead, we explored the model’s ability to detect and classify RB in a multiracial cohort. Despite unequal frequency distribution between the races, we identified the scope for improvement and retrained the AI model to detect and classify RB. The AI model displayed an accuracy of 97% for detecting RB and 98%, 93%, >99%, 94%, and 93% for grouping tumors into the International Classification of Retinoblastoma groups A to E, respectively.

**Abstract:**

**Background:** The color variation in fundus images from differences in melanin concentrations across races can affect the accuracy of artificial intelligence and machine learning (AI/ML) models. Hence, we studied the performance of our AI model (with proven efficacy in an Asian-Indian cohort) in a multiracial cohort for detecting and classifying intraocular RB (iRB). **Methods:** Retrospective observational study. **Results:** Of 210 eyes, 153 (73%) belonged to White, 37 (18%) to African American, 9 (4%) to Asian, 6 (3%) to Hispanic races, based on the U.S. Office of Management and Budget’s Statistical Policy Directive No.15 and 5 (2%) had no reported race. Of the 2473 images in 210 eyes, 427 had no tumor, and 2046 had iRB. After training the AI model based on race, the sensitivity and specificity for detection of RB in 2473 images were 93% and 96%, respectively. The sensitivity and specificity of the AI model were 74% and 100% for group A; 88% and 96% for group B; 88% and 100% for group C; 73% and 98% for group D, and 100% and 92% for group E, respectively. **Conclusions:** The AI models built on a single race do not work well for other races. When retrained for different races, our model exhibited high sensitivity and specificity in detecting RB and classifying RB.

## 1. Introduction

Retinoblastoma (RB) is a relatively new medical frontier that artificial intelligence (AI) has touched. The earliest literature evidence of artificial intelligence in the field of RB was in 2017 by Ciller C et al., who demonstrated the use of AI for the longitudinal tracking of RB tumors [1]. While this opened doors to several possibilities for AI’s application in RB, the primary focus has been detecting RB rather than follow-up [2,3,4,5,6,7,8,9,10,11,12]. This is probably the right direction since early detection of RB is a bigger problem worldwide than monitoring response to therapy, especially in middle and low-income countries [13,14]. Only a few groups, including ours, have trained and tested the AI and machine learning (ML) models for detecting RB from RetCam^TM^/fundus images of patients from their respective centers [6,10,11,12]. In contrast to other studies that focused on the detection of RB in any given fundus image [1,10], we have looked into the accuracy of detection of smaller tumors, i.e., tumors classified as group A and B based on the International Classification of Retinoblastoma (ICRB) [15]. We noted that the sensitivity for detecting group A tumors ranged from 56% to 100%, and group B tumors ranged from 82% to 95% in eyes with RB, identifying a definite scope for improvement [11]. The accuracy of the AI model can be improved by retraining and testing the AI model with larger volumes of high-quality data that exhibit adequate diversity and granularity [16]. With this goal, we undertook the present study in a large and diverse multiracial image database, and herein we report the results.

## 2. Methods

### 2.1. Data Collection

This was a retrospective study that adhered to the tenets of the Declaration of Helsinki and was compliant with the U.S. Health Insurance Portability and Accountability Act norms. The Institutional Review Boards approved the study at LV Prasad Eye Institute, Hyderabad, India, and the Wills Eye Hospital, Philadelphia, U.S.A. Fundus images at presentation were retrieved for patients with RB availing the Ocular Oncology Service, Wills Eye Hospital, Philadelphia, USA, between July 2022 and June 2023. The only data obtained besides the images was the patient’s race since we anticipated a change in the existing AI model’s ability to detect RB due to heterogeneity in the fundus pigmentation in different races [17,18,19,20,21]. The information about the patient’s race was retrieved from the medical records as reported by the parent or caregiver. Per institute protocols at the Ocular Oncology Service, Wills Eye Hospital, Philadelphia, U.S.A, race was recorded as White, Hispanic, African American, Asian Indian, Asian Oriental, or Others.

As of March 2024, the Statistical Policy Directive no. 15 on standards for maintaining, collecting, and presenting federal data on race and ethnicity combined race and ethnicity into one single question and listed the following categories: American Indian or Alaska Native, Asian, Black or African American, Hispanic or Latino, Middle Eastern or North African, Native Hawaiian or Pacific Islander, and White [22,23]. We largely adhered to this format. Although “Hispanic” is considered more appropriate as an ethnonym than a race, this cohort was deemed distinct to the study since their fundus pigmentation is known to vary from the White population [22,23,24,25]. Hispanics included individuals of Mexican, Puerto Rican, Salvadoran, Cuban, Dominican, Guatemalan, and other Central or South American or Spanish culture or origin [22]. We included consecutive patients of all races, and heterogeneity was expected in the number of patients of different races, given the predominance of White patients seeking services at Wills Eye Hospital, Philadelphia, U.S.A. No identifiable information was retrieved, and the Health Insurance Portability and Accountability Act was adhered to. Poor-quality images were excluded from the database. Individual images and the respective eyes were assigned an ICRB group label (A to E) [16], or as ‘normal’ by two independent ocular oncologists (VSV, KB). The degree of interobserver variability was assessed for images and eyes.

### 2.2. Feature Extraction

Our previous work elaborates on the details of the methods used for feature extraction [11,12]. Broadly, a combination of the MultiLabel Classification Model of Deep Learning, Mobile-Net v2 SSD-based deep learning model for transfer learning, Computer Vision techniques, and geometric calculations was used. Application of our pre-existing algorithm with multiple AI/ML and vision models that were trained on fundus images of Asian Indian eyes [12] performed poorly for a multiracial cohort.

### 2.3. Identification of Specific Problems and Development of Solutions

Figure 1 illustrates the methodology employed. The overview’s upper section (Figure 1, above the dotted line) describes steps in extracting various features of the images using deep learning (AI) and computer vision models. The overview’s lower section (Figure 1, below the dotted line) contains an AI/ML classifier model for the final diagnosis. As anticipated, we noted that the problems with the AI model’s interpretation were primarily due to differences in fundus color due to variable melanin based on race, contrasting blood vessels and hemorrhage with respect to the fundus, and subtle feature interactions for the classification of RB. Detailed analysis pointed towards the need for changes in the AI model to detect optic discs, tumors, blood vessels, and hemorrhage (Figure 1, marked in red). 

With these changes, while the accuracy detection of RB was acceptable, we learned that further retraining was needed to improve the accuracy of the classification of RB. We further elaborate on the changes made to the previously validated model here:Fundus color: This significantly impacted the AI/ML models for detecting optic discs and tumors across various races. While the larger tumors did not pose much of a problem, the smaller ones, especially groups A and B, often blended within the fundus color (Figure 2). This problem was mitigated by retraining, testing, and validating against the multiracial data, thus ensuring no data imbalance across different categories.Fundus contrast: The fundus images for Asian-Indian and African American races had a clear contrast between blood vessels, hemorrhages, and the background fundus color. The fundus images did not have such a contrast for the White race. Our default Computer Vision feature extraction model from the pre-existing algorithm proved too broad, resulting in incorrect grouping to C, D, and E groups. Sometimes, the AI model assigned group D or E labels to group A tumors. To bring about a more versatile broader algorithm applicable to all races, we appended a pre-processing step to perform color segmentation and a set of enhancements. The color segmentation and contrast enhancement approaches were used only to extract features such as blood vessels and hemorrhages. We retrained the models for other features to capture other variabilities, as a standard practice described in the literature [26]. This pre-processing step transformed the cohort with multiracial fundus images into a common framework so that the blood vessel and hemorrhage detection computer models could be uniformly applied. Figure 3 illustrates the three-step process we employed: pre-processing, identification of blood vessels/hemorrhage, and computing features for classification, such as the area covered and adjacent retina visible or not visible.Retraining of the AI model: Due to the differences in fundus color and contrast, the AI model showed a propensity towards assigning inaccurate C, D, and E group labels in the multiracial dataset. To delineate and enable the model to learn this subtle difference from the data set, we re-trained the AI model using a new ML model on the features extracted from various new/modified models, as described in Figure 1 and Figure 2. A total of 14 extracted and derived features were used to build an XGBoost ML model for grouping RB.Assignment of a group label to an eye: Since different images of the same eye would show variation in the presence or absence of the tumor, its size, and associated features, variation was expected for group labels assigned to the images. Several factors were carefully considered when assigning a group label to the eye based on group labels of individual images. After grouping every image of an eye to arrive at a patient-level diagnosis, we followed a two-step process, which we termed *aggregation*. The first step in the aggregation was to remove outliers. We noted that within the large dataset, there were subtle inadequacies, such as a lack of focus only on a small set of images of an eye. Instead of invalidating such eyes and limiting the analysis, the outlier step in our aggregation algorithm removed these if (i) there were ≥5 such images in an eye and (ii) there was only one image with a group assigned ≥2 groups away from the rest of the images of that eye. The second step in assigning a group label to the eye was to look for the most frequent group label assigned within the images of that eye that is greater than a threshold of 65% or defaults to a conservative stance of the highest group among the images if there is no clear majority.

### 2.4. Performance Metrics and Statistical Analysis

The overall accuracy of the AI model, as well as misclassification, under-classification, and over-classification rates, were calculated. The sensitivity, specificity, positive, and negative predictive values were calculated for the entire cohort based on the ICRB group and race.

## 3. Results

### 3.1. Dataset, Groups, and Distribution

The study included 2473 images from 210 eyes. The eyes were classified as ICRB group A (*n* = 140; 6%), group B (*n* = 876; 35%), group C (*n* = 92; 4%), group D (*n* = 556; 22%), group E (*n* = 382; 15%), and normal (*n* = 427; 17%) by VSV and KB. The interobserver variability was <1% for images and <1% for eyes. The distribution of images and eyes of the dataset between different ICRB groups and races is listed in Table 1. A majority of the eyes belonged to group B and the White race. The distribution of images used for training, validation, and testing are tabulated in Table 2.

### 3.2. Performance Metrics

The sensitivity, specificity, positive predictive value, and negative predictive value for detection of RB in 2473 images were 93%, 96%, 99%, and 74%, respectively, and in 210 eyes were 98%, 96%, 99%, and 92%, respectively. The sensitivity, specificity, positive predictive value, and negative predictive value of the AI model were 74%, 100%, 100%, and 97% for group A; 88%, 96%, 91%, and 94% for group B; 88%, 100%, 100%, and >99% for group C; 73%, 98%, 90%, and 94% for group D, and 100%, 92%, 67%, and 100% for group E, respectively. The performance metrics for images are tabulated alongside the metrics for eyes in Table 3.

The accuracy for detecting RB was 94% for images. The under-classification rate (including 142 images that were misclassified as ‘normal’) for 2473 images was 11% (*n* = 219), and the over-classification rate was 5% (*n* = 96). Regarding the assignment of ICRB group labels, 84% were correctly assigned overall, 90% within one ICRB group difference, and 98% within two ICRB group differences. These accuracies improved at an eye level, which was expected. The accuracy for detecting RB was 97% for eyes. The under-classification rate (including 4 eyes that were misclassified as ‘normal’) for 210 eyes was 3% (*n* = 5), and the over-classification rate was 12% (*n* = 20). With respect to the ICRB group labels, 87% were assigned the correct group overall, 94% were within one ICRB group difference, and 98% were within two ICRB group differences. The comparison of the ICRB group assigned by the AI model versus that assigned by the ocular oncologists is depicted as matrices for the images and eyes in Figure 4.

### 3.3. Analysis of the Sensitivity for Detecting RB by the AI Model Based on the Race and ICRB Group

Table 4 lists the AI model’s sensitivities within each racial cohort, further subclassified by the ICRB group. Overall, the sensitivity was highest at 100% for the Asian Indian and Asian Oriental cohorts, 86% for the White, and 89% for the African American cohorts.

Overall, the AI model detected and classified RB in diverse races and all ICRB groups (Figure 5).

## 4. Discussion

Large-volume and high-quality data are essential for effectively training AI models [16]. This is also expected to improve over time as datasets enlarge and become more diverse. An excellent example is the improvement in sensitivity and specificity of the AI models used to detect ROP [27,28]. Early studies dating back to 2002 with fundus images, 11 ROP and 9 non-ROP participants showed 82% and 75% sensitivity and specificity, respectively [29]. However, with a dataset of 227,326 retinal images, in 2023, the sensitivity and specificity reached 91% and 91%, respectively [30]. This trend will likely continue for RB if the AI model is trained with expanded datasets. 

The major leap in this work from our previous reports is the improvement in the detection of ICRB group A tumors from 56% to 74%. While other studies report an accuracy of 82% to 100% based on various models, the data on the AI model’s efficacy for tumors of different ICRB groups was limited. In a study by Zhang et al., with a reported overall accuracy of 99% for detecting RB, the sensitivity was 66% for eyes with group A tumors [10]. 

The second important aspect of this work is understanding race’s impact on the AI model’s efficacy in detecting RB. For an AI model to function as an efficient RB screening tool and be utilizable globally, it should provide reliable results irrespective of racial and ethnic differences from good-quality fundus images. The background color in fundus images can vary significantly based on the amount of melanin present in the retinal pigment epithelium and choroid [17,18,19,20,21]. Even the density of macular pigment and vessel architecture varies between cohorts of different ethnicities [18,19]. The color of a target object and its background can significantly affect the accuracy of AI models, and this has been shown with the detection of objects such as medicine pills [31] and in real-time fundus images [32]. Thus, the background fundus color is a crucial factor that needs to be considered while employing AI-based tools for early detection of RB in a community setting. 

Coyner et al. described the ability of AI models to “learn” the ethnicity from fundus images, which leads to a bias in the interpretation of images of newborns screened for retinopathy of prematurity [32]. The findings in our study were in contrast to the observations by Coyner et al. [32], wherein the AI model trained in Asian Indian eyes did not perform as predicted with eyes of White origin before retraining. Nevertheless, in both scenarios, the AI model’s output was affected by race, and to improve accuracy, the AI models need to be trained from diverse cohorts. Some groups have attempted to use AI to predict the degree of fundus pigmentation and guide additional training on the AI model [17]. Since this was not the primary problem we aimed to solve with AI, we employed an alternate approach that seemed to overcome this drawback and improve the AI model’s accuracy. 

Thus, in this study, we saw that the accuracy of the AI model varied between different races, which warranted retraining. Promising results were obtained after retraining the AI model for all races. Another option to improve the accuracy is to develop other models for various features and a separate AI model for each race. We deliberately avoided this for the following reasons.

We developed a generic model that can be easily deployed across a clinical practice with multiracial data, avoiding the need for multiple models. A single algorithm with one set of models will be versatile in deployment, retraining, and testing.Further, prior definitive information on the race may only sometimes be available to apply an appropriate model based on that information. Instead, selecting a common model for multiracial data agnostic to the race is more versatile. We measured the performance against various races and verified that our proposed approach worked, proving that the learning of different AI/ML models is uniform.

This study’s diverse database enabled our AI model to improve its accuracy in detecting smaller tumors. It brought to light an important aspect that affects the accuracy of the AI model in detecting and grouping RB, which is the race of affected children with RB. Nevertheless, further training with eyes of different origins is needed before it can be translated into a universally deployable RB screening technology.

Being one of the initial studies in this arena, this study has limitations that could not be eliminated. The primary goal was to improve the AI model’s accuracy for small tumors by training it on a large and diverse cohort. Firstly, the distribution of images from different races was not uniform. Because the study was based on data from a single Ocular Oncology Center in the United States, most of the patients were White [23]. Unequal distribution in the different cohorts may have affected the performance of the AI model for each race. Thus, this AI model‘s generalizability beyond Asian Indians and Whites is still limited. On the same lines, whether a screening program based on this model needs to be race-specific or universal cannot be answered yet. Multicentric data-sharing studies are warranted to bridge these lacunae, especially for underrepresented groups and groups not represented in this study. However, challenges to overcome on this front would be acquiring uniformly good quality. The second limitation of this study was that despite acquiring a large cohort of images, the numbers in ICRB Groups A and B were smaller. To employ this technology as a screening tool needs improvement and further training on these sub-cohorts. Lastly, a few features may have been missed by the combination of approaches we used to train the model, affecting the model’s accuracy, and this can be improved through further studies over time. 

## 5. Conclusions

This is a first-of-its-kind study to explore the ability of an AI model to detect and classify RB in a multiracial cohort. As we advance, multicentric global collaborations and data sharing can enhance the AI model’s accuracy for detecting RB in different global cohorts and smaller tumors.

## Figures and Tables

**Figure 1 cancers-16-03516-f001:**
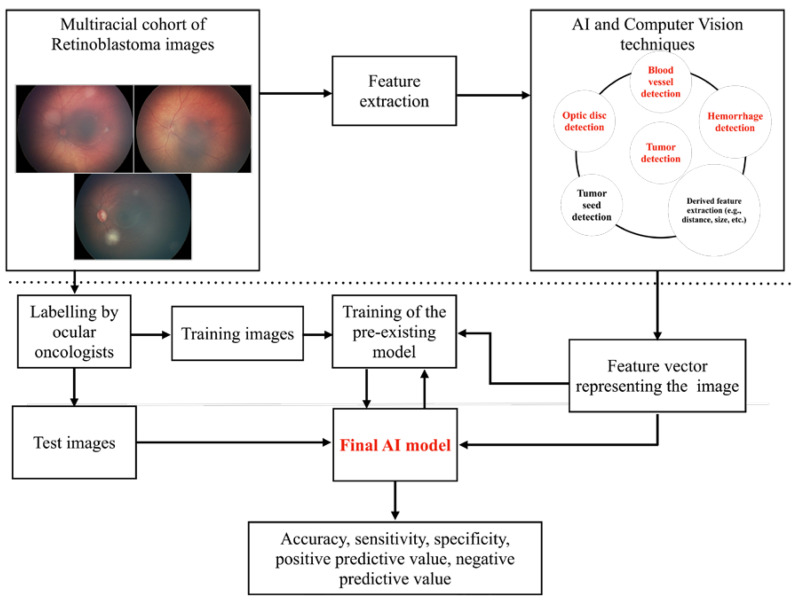
Overview of the methodology employed.

**Figure 2 cancers-16-03516-f002:**
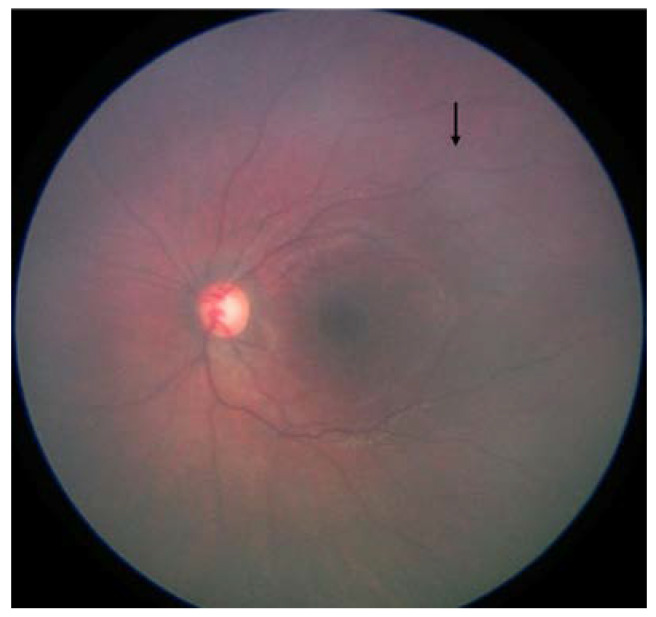
Fundus photograph with subtle retinoblastoma (arrow).

**Figure 3 cancers-16-03516-f003:**
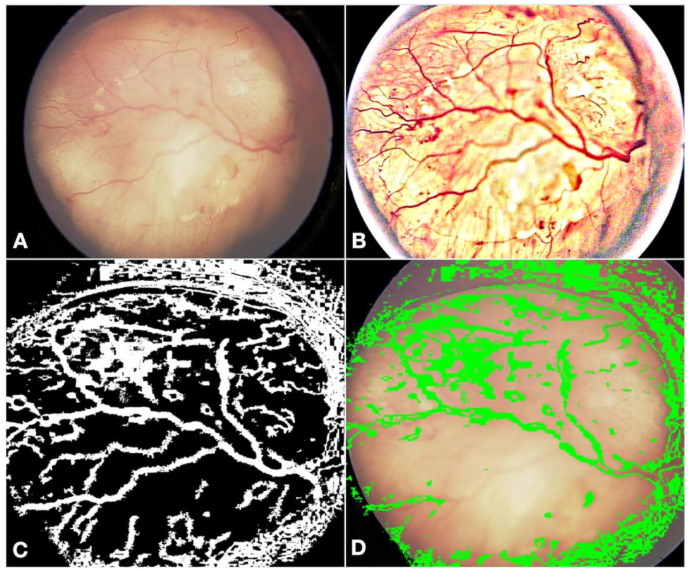
The three-step process employed in image processing. Illustration of the original image (**A**) that was pre-processed (**B**) to ensure uniformity in the identification of blood vessels/hemorrhages across races (**C**). Further, features such as the area covered by the tumor (**D**) were computed for grouping.

**Figure 4 cancers-16-03516-f004:**
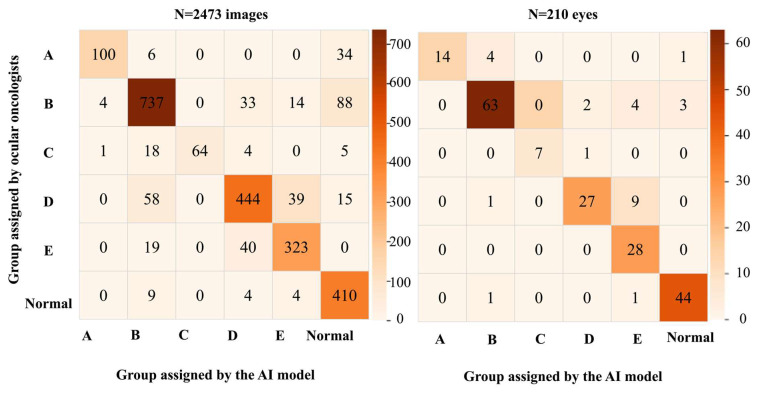
Performance metrics of the AI model for detection and classification of RB depicted as matrices for 2473 images and 210 eyes.

**Figure 5 cancers-16-03516-f005:**
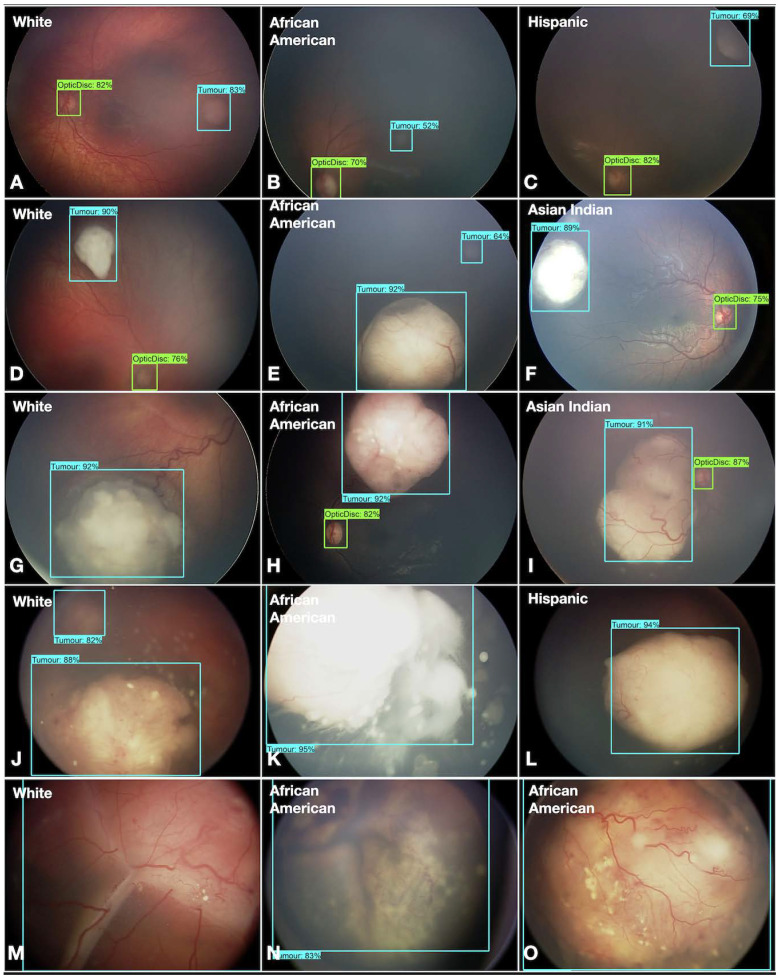
The AI model. The AI model detects optic disc (green bounding box with % confidence) and retinoblastoma (blue bounding box with % confidence) in ICRB groups A (**A**–**C**), B (**D**–**F**), C (**G**–**I**), D (**J**–**L**), and E (**M**–**O**) in White, African American, Hispanic, and Asian-Indian races.

**Table 1 cancers-16-03516-t001:** Artificial intelligence and machine learning in retinoblastoma: Distribution of images and eyes used for training, validation, and test.

Learning Model	Number of Images Labelled for Training + Validation + Test of the AI Model (*n*)	Number of Images Used for Training (*n*)	Number of Images Used for Cross-Validation (*n*)	Number of Images Used for Test (*n*)
Deep learning (MultiLabel Classification: optic disc + tumor)	784	501	126	157
Deep Learning (seeds)	132	84	21	27
Final AI model (XGBoost model)	2473	1582	396	495

**Table 2 cancers-16-03516-t002:** Artificial intelligence and machine learning in retinoblastoma: Distribution of 2473 images of 210 eyes by ICRB group and race.

	No. of Images							No. of Eyes						
	Race							Race						
ICRB Group	W	AA	H	AI	AO	NA	All Races	W	AA	H	AI	AO	NA	All Races
Normal	328	72	13	5	6	3	427	31	10	2	1	1	1	46
A	120	17	1	0	0	2	140	16	2	0	0	0	1	19
B	705	125	22	15	0	9	876	57	9	1	2	0	3	72
C	70	6	0	16	0	0	92	5	1	0	2	0	0	8
D	394	118	28	13	3	0	556	25	8	2	1	1	0	37
E	276	84	13	0	9	0	382	19	7	1	0	1	0	28
All groups	1893	422	77	49	18	14	2473	153	37	6	6	3	5	210

W = White; AA = African American; H = Hispanic; AI = Asian Indian; AO = Asian Oriental; NA = not available.

**Table 3 cancers-16-03516-t003:** Artificial intelligence and machine learning in retinoblastoma: Overall performance metrics of the final AI model in 2473 images of 210 eyes.

	Detection of RB		Normal		A		B		C		D		E	
	N = 2473	n = 210	N = 427	n = 46	N = 140	n = 19	N = 876	n = 72	N = 92	n = 8	N = 556	n = 37	N = 382	n = 28
Ac	94%	97%	94%	97%	98%	98%	90%	93%	99%	>99%	92%	94%	95%	93%
Sn	93%	98%	96%	96%	71%	74%	84%	88%	70%	88%	80%	73%	85%	100%
Sp	96%	96%	93%	98%	>99%	100%	93%	96%	100%	100%	96%	98%	97%	92%
PPV	99%	99%	74%	92%	95%	100%	87%	91%	100%	100%	85%	90%	85%	67%
NPV	74%	92%	99%	99%	98%	97%	91%	94%	99%	>99%	94%	94%	97%	100%

N = no. of images; n = no. of eyes; Ac = accuracy, Sn = sensitivity; Sp = specificity; PPV = positive predictive value; NPV = negative predictive value.

**Table 4 cancers-16-03516-t004:** Artificial intelligence and machine learning in retinoblastoma: Sensitivity of the AI model based on International Classification of Retinoblastoma (ICRB) group and race in 210 eyes.

Race	Normal (*n*)	A (*n*)	B (*n*)	C (*n*)	D (*n*)	E (*n*)	Total (*n*)
White (*n* = 153)	94% (31)	75% (16)	88% (57)	80% (5)	68% (25)	100% (19)	86% (153)
African American (*n* = 37)	100% (10)	50% (2)	89% (9)	100% (1)	75% (8)	100% (7)	89% (37)
Asian Indian (*n* = 6)	100% (1)	na	100% (2)	100% (2)	100% (1)	na	100% (6)
Asian Oriental (*n* = 3)	100% (1)	na	na	na	100% (1)	100% (1)	100% (3)
Hispanic (*n* = 6)	100% (2)	na	100% (1)	na	100% (2)	100% (1)	100% (6)
Not available (*n* = 5)	100% (1)	100% (1)	67% (3)	na	Na	na	80% (5)
Total (*n* = 210)	96% (46)	74% (19)	88% (72)	88% (8)	73% (37)	100% (28)	87% (210)

na = sensitivity could not be computed since there were no eyes in that sub-cohort.

## Data Availability

Data is available upon request.

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
