# Peer review of "Artificial Intelligence and Machine Learning in Ocular Oncology, Retinoblastoma (ArMOR): Experience with a Multiracial Cohort"

_cancers, 2024, doi:10.3390/cancers16203516_

Round 1

Reviewer 1 Report

Comments and Suggestions for Authors

The article titled "Artificial Intelligence and Machine Learning in Ocular Oncology, Retinoblastoma (ArMOR), Paper III: Experience with a Multiracial Cohort" discusses the use of artificial intelligence (AI) and machine learning (ML) for detecting and classifying retinoblastoma (RB) in a multiracial cohort. Retinoblastoma is the most common eye cancer in childhood, and early detection is crucial for improving outcomes. The study utilized an AI model previously trained on an Indian cohort to detect and classify RB in a multiracial cohort comprising White, African-American, Asian, and Hispanic individuals.

The results showed that variations in fundus color, caused by differences in melanin concentration across races, affected the AI model's accuracy. After retraining the model with data from different races, the accuracy for detecting RB was 93%, with a specificity of 96%. The study concludes that AI models built for a single race do not work well for other races and that training on multiracial data significantly improves the model's accuracy.

Major Review:

The study acknowledges that the distribution of patients among different races was not uniform, which may have affected the results and generalization of the model. A review should include a more balanced sample or discuss this limitation's impact in more detail.

While the article mentions improvements in detecting smaller tumors after retraining, accuracy is still below ideal for groups A and B. This underscores the urgent need for further research and development in AI and ML to improve sensitivity in these critical cases.

The model's performance varies significantly among racial groups, with some exhibiting very low sensitivities. The review should address whether a genuinely global model can be developed or whether race-specific models would be more effective.

The cohort studied is limited to a specific institution and a restricted multiracial cohort. The review could suggest the need for multicenter studies to ensure the generalization of results to other global populations.

The color segmentation and contrast enhancement approach to standardize fundus images across races may only capture some variabilities. A review could investigate alternative or improved methodologies.

The article mentions the importance of high-quality and large-volume data for training effective AI models. This underscores the significant impact of these factors on the results and the need for a deeper exploration of their influence. It is recommended to discuss more deeply the impact of data quality and volume and consider including a sensitivity analysis.

Comments on the Quality of English Language

None

Author Response

1. The article titled "Artificial Intelligence and Machine Learning in Ocular Oncology, Retinoblastoma (ArMOR), Paper III: Experience with a Multiracial Cohort" discusses the use of artificial intelligence (AI) and machine learning (ML) for detecting and classifying retinoblastoma (RB) in a multiracial cohort. Retinoblastoma is the most common eye cancer in childhood, and early detection is crucial for improving outcomes. The study utilized an AI model previously trained on an Indian cohort to detect and classify RB in a multiracial cohort comprising White, African-American, Asian, and Hispanic individuals.

The results showed that variations in fundus color, caused by differences in melanin concentration across races, affected the AI model's accuracy. After retraining the model with data from different races, the accuracy for detecting RB was 93%, with a specificity of 96%. The study concludes that AI models built for a single race do not work well for other races and that training on multiracial data significantly improves the model's accuracy.

Response: Thank you for the comments 

Comment

Response

Changes

Location

1. The study acknowledges that the distribution of patients among different races was not uniform, which may have affected the results and generalization of the model. A review should include a more balanced sample or discuss this limitation's impact in more detail.

Thank you for this suggestion. We have added this information to limitations.

“Being one of the initial studies in this arena, this study has limitations that could not be eliminated. The primary goal was to improve the AI model’s accuracy for small tumors by training it on a large and diverse cohort. Firstly, the distribution of images from different races was not uniform. Because the study was based on data from a sin-gle Ocular Oncology Center in the United States, most of the patients were White [23]. Unequal distribution in the different cohorts may have affected the performance of the AI model for each race. Thus, this AI model‘s generalizability beyond Asian Indians and Whites is still limited. On the same lines, whether a screening program based on this model needs to be race-specific or universal cannot be answered yet. Multicentric data-sharing studies are warranted to bridge these lacunae, especially for underrepre-sented groups and groups not represented in this study. However, challenges to over-come on this front would be acquiring uniformly good quality”

Page 10, Discussion

2. While the article mentions improvements in detecting smaller tumors after retraining, accuracy is still below ideal for groups A and B. This underscores the urgent need for further research and development in AI and ML to improve sensitivity in these critical cases.

Thank you for this comment. We have acknowledged this limitation and discussed the need for further work on this front.

“The second limitation of this study was that despite acquiring a large cohort of images, the numbers in ICRB Groups A and B were smaller. To employ this technology as a screening tool needs improvement and further training on these sub-cohorts.”

Page 10, Discussion

3. The model's performance varies significantly among racial groups, with some exhibiting very low sensitivities. The review should address whether a genuinely global model can be developed or whether race-specific models would be more effective.

Thank you for allowing us to clarify this. At this point it is difficult to comment on the effectiveness of a race-specific model versus a universally applicable model given the heterogeneity of data available. We have discussed this in limitations and suggest the future direction of work.

“On the same lines, whether a screening program based on this model needs to be race-specific or universal cannot be answered yet. Multicentric data-sharing studies are war-ranted to bridge these lacunae, especially for underrepresented groups and groups not represented in this study.”

Page 10, Discussion

4. The cohort studied is limited to a specific institution and a restricted multiracial cohort. The review could suggest the need for multicenter studies to ensure the generalization of results to other global populations.

Thank you for this comment. We have acknowledged this limitation.

“Because the study was based on data from a single Ocular Oncology Center in the United States, most of the patients were White [23]. Unequal distribution in the differ-ent cohorts may have affected the performance of the AI model for each race.”

Page 10, Discussion

5. The color segmentation and contrast enhancement approach to standardize fundus images across races may only capture some variabilities. A review could investigate alternative or improved methodologies.

Thank you for allowing us to add to this. The color segmentation and contrast enhancement approaches were used only for extracting features such as blood vessels and haemorrhages and we did not see that much variability in extracting these features. For other features, we retrained the models to capture the variabilities.

However, we do agree that some features may have been missed by the approach employed and added this to limitations.

“Lastly, a few features may have been missed by the combination of approaches we used to train the model, affecting the model’s accuracy, and this can improve through further studies over time.”

Page 10, Discussion

6. The article mentions the importance of high-quality and large-volume data for training effective AI models. This underscores the significant impact of these factors on the results and the need for a deeper exploration of their influence. It is recommended to discuss more deeply the impact of data quality and volume and consider including a sensitivity analysis.

Thank you for this valuable insight. It is indeed a known fact that data quality and volumes impact the accuracy of an AI model and we have added the literature relevant to this.

“Large-volume and high-quality data are essential for effectively training AI mod-els [16, 27]. This is also expected to improve over time as datasets enlarge and become more diverse. An excellent example is the improvement in sensitivity and specificity of the AI models used to detect ROP [28, 29]. Early studies dating back to 2002 with fun-dus images, 11 ROP and 9 non-ROP participants showed 82% and 75% sensitivity and specificity, respectively [30]. However, with a dataset of 227,326 retinal images, in 2023, the sensitivity and specificity reached 91% and 91%, respectively [31]. This trend will likely continue for RB if the AI model is trained with expanded datasets.”

Page 9, Discussion

Reviewer 2 Report

Comments and Suggestions for Authors

The authors presented a paper about “Artificial Intelligence and Machine Learning in Ocular

Oncology, Retinoblastoma (ArMOR), Paper III: Experience with a Multiracial Cohort”.

The topic of Artificial intelligence has been largely debated in the recent literature but the evidence with specific regard to ocular oncology is scanty.

The author claim that the workflow used by the authors has already been described in their previous articles but as far as I can see for example reference 12 referes to unpublished data so I believe that the title should not be "III" since the second paper has not been published yet  (Kaliki, S.; Vempuluru, V.S.; Patil, G.; Viriyala, R.; Dhara, K.K. Artificial intelligence and machine learning in ocular oncology, retinoblastoma (ArMOR): Paper II [Unpublished data]).

I have some doubts about the classification used to identify White, African-American, Asian, and Hispanic patients: is there a specific scientific reference to frame patients according to this method?

For example what about the label of “white” patient: could Caucasian be used as alternative? Again regarding “Hispanic” even though I believe it is easily possible from a cultural point of view to use this label I am not quite convinced that this could be used in a scientific article: which countries would be included under this definition?

Please provide more robust scientific evidence about the choice to label patients under a certain racial category and the methodology followed to do so.  

Author Response

The authors presented a paper about “Artificial Intelligence and Machine Learning in Ocular

Oncology, Retinoblastoma (ArMOR), Paper III: Experience with a Multiracial Cohort”.

The topic of Artificial intelligence has been largely debated in the recent literature but the evidence with specific regard to ocular oncology is scanty.

Response: Thank you for the comments

Comment

Response

Changes

Location

Initial comments

1. The author claim that the workflow used by the authors has already been described in their previous articles but as far as I can see for example reference 12 referes to unpublished data so I believe that the title should not be "III" since the second paper has not been published yet (Kaliki, S.; Vempuluru, V.S.; Patil, G.; Viriyala, R.; Dhara, K.K. Artificial intelligence and machine learning in ocular oncology, retinoblastoma (ArMOR): Paper II [Unpublished data]).

Thank you for this input. We have deleted citation of this work.

We have deleted: Kaliki, S.; Vempuluru, V.S.; Patil, G.; Viriyala, R.; Dhara, K.K. Artificial intelligence and machine learning in ocular oncology, retinoblastoma (ArMOR): Paper II [Unpublished data]

Page 11, References

2. I have some doubts about the classification used to identify White, African-American, Asian, and Hispanic patients: is there a specific scientific reference to frame patients according to this method?

Thank you for giving us the opportunity to clarify this. The information of race was parent-reported.

We have added this information, “The information about the patient’s race was retrieved from the medical records as re-ported by the parent or caregiver. Per institute protocols at the Ocular Oncology Ser-vice, Wills Eye Hospital, Philadelphia, U.S.A, race was recorded as White, Hispanic, African American, Asian Indian, Asian Oriental, or Others.”

Page 2, Methods

3. For example what about the label of “white” patient: could Caucasian be used as alternative?

Thank you for this suggestion. However, we adhered to the terminology prescribed by the United States Census Bureau in adherence to the 1997 Office of Management and Budget and hence retain the term, “White”

We have added this information, “As of March 2024, the Statistical Policy Directive no. 15 on standards for maintain-ing, collecting, and presenting federal data on race and ethnicity combined race and ethnicity into one single question and listed the following categories: American Indian or Alaska Native, Asian, Black or African American, Hispanic or Latino, Middle Eastern or North African, Native Hawaiian or Pacific Islander, and White [22, 23].”

Page 2, Methods

Again regarding “Hispanic” even though I believe it is easily possible from a cultural point of view to use this label I am not quite convinced that this could be used in a scientific article: which countries would be included under this definition?

We agree with the reviewer that “Hispanic”is more appropriately an ethnonym.2,3 However, we distinguished this cohort from the Caucasians due to differences in the fundus pigmentation.4,5

We clarified this, “We largely adhered to this format. Although “Hispanic” is considered more appropriate as an ethnonym than a race, this cohort was deemed distinct to the study since their fundus pigmentation is known to vary from the White population [22-25].”

Page 2, Methods

Please provide more robust scientific evidence about the choice to label patients under a certain racial category and the methodology followed to do so.

Thank you for this recommendation. We have added this information in the Methods section.

We have clarified this better, “As of March 2024, the Statistical Policy Directive no. 15 on standards for maintain-ing, collecting, and presenting federal data on race and ethnicity combined race and ethnicity into one single question and listed the following categories: American Indian or Alaska Native, Asian, Black or African American, Hispanic or Latino, Middle Eastern or North African, Native Hawaiian or Pacific Islander, and White [22, 23]. We largely adhered to this format. Although “Hispanic” is considered more appropriate as an eth-nonym than a race, this cohort was deemed distinct to the study since their fundus pigmentation is known to vary from the White population [22-25]. We included consec-utive patients of all races, and heterogeneity was expected in the number of patients of different races, given the predominance of White patients seeking services at Wills Eye Hospital, Philadelphia, U.S.A.”

Page 2, Methods

Reviewer 3 Report

Comments and Suggestions for Authors

See attached file

Comments on the Quality of English Language

No comment

Author Response

This is a retrospective monocentric work focused on the improvement of a model of artificial intelligence (AI) and machine learning able to detect and classify with high sensivity Retinoblastoma (RB) in a multiracial cohort. Its main contribute is represented by the aim to employ the AI to a relatively new medical frontier of the RB field. In fact, an AI model able to detect and classify RB with high sensitivity could become an efficient RB screening tool usable globally. The AI model displayed an high accuracy of 97% for the detection of RB and the grouping of tumors in International Classification of Retinoblastoma; moreover there is an improvement in the detection of ICRb group A tumor with a sensitivity that reaches the 74% so the AI model could help the RB experts in the diagnosis of the smallest tumors.

Response: Thank you for the comments 

Comment

Response

Changes

Location

1. The categorization of the eyes based on the race, if on one hand is a limit – in fact the AI model accuracy is different for every race since the background fundus color is a crucial factor that needs to be considered while employing AI-based tools – on the other hand it could perform an algorithm able to “learn” the ethnicity from fundus images; so to make easier and faster the diagnosis.

The title is generic and it suggests that the applied analysis is homogeneous for all ethnic group. The summary speaks in general about the purpose of the paper, that is, to explore a different cohort at the racial level and for this reason the study is expected to be neutral and generalizable.

Thank you for this input. We agree with the reviewer’s observation. We have acknowledged this the frequency distribution of images from different races as a limitation.

“Firstly, the distribution of images from different races was not uniform. Because the study was based on data from a single Ocular Oncology Center in the United States, most of the patients were White [23].”

Page 10, Discussion

In the abstract data evidence a large discrepancy between the white ethnicity and all the others. There is a too strong mathematical difference to generalise the study, precisely because it is supposed to be applicable to different races.

“we explored the model’s ability to detect and classify RB in a multiracial cohort, identified the scope

for improvement, and retrained the AI model to detect and classify RB in a multiracial cohort.”

This sentence seems to be talking about a multiracial cohort that is perfectly various, therefore the

study seems to be generalizable.

We appreciate the reviewers input in this regard. We agree that the AI model is not practically generalizable to different races at this stage and have acknowledged this as a limitation. However, through our observations we note that this is in an important factor that affects the model’s performance and detection of RB. However, we believe this study is an initial step which which eventually improve with time and volumes into a more generalizable model.

“On the same lines, whether a screening program based on this model needs to be race-specific or universal cannot be answered yet. Multicentric data-sharing studies are war-ranted to bridge these lacunae, especially for underrepresented groups and groups not represented in this study.”

Page 10, Discussion

“Results: Of 210 eyes, 153 (73%) belonged to White, 37 (18%) to African-American, 9 (4%) to Asian,

6 (3%) to Hispanic patients, and 5 (2%) had no reported race.”

This clarification regarding the ethnic proportions in the study makes the previous observations contradictory, highlighting how the general applicability of the study can be affected by the numerical disparity at the ethnic level.

Thank you for this input. We have acknowledged that the frequency distribution of images from different races as a limitation.

“Firstly, the distribution of images from different races was not uniform. Because the study was based on data from a single Ocular Oncology Center in the United States, most of the patients were White [23].”

Page 10, Discussion

“The only data obtained besides the images was the patient’s race.”

This sentence reported within the data collection highlights once again how the ethnic aspect is central to the study. Therefore, having chosen such a high number of white patients continues to seem very counterproductive.

Thank you for allowing us to clarify this. Images from all patients within the study period were included. The number of images retrived was not based on race per se and we have added this to limitations. We also mention that these under-represented cohorts should be studied further to make the AI model more generalisable.

“Because the study was based on data from a single Ocular Oncology Center in the United States, most of the patients were White [23]. Unequal distribution in the differ-ent cohorts may have affected the performance of the AI model for each race.”

Page 10, Discussion

“Table 4 lists the AI model's sensitivities within each racial cohort, further subclassified by the ICRB

group. Overall, the sensitivity was highest at 100% for the Asian Indian and Asian Oriental cohorts,

and it reached 86% for the White cohort and 89% for the Afri can American cohorts.”

From these data, a doubt arises: are the sensitivity values for each race influenced by the numerical

disparity between the various races within the multi-ethnic sample? If so, one would have to

consider whether a more balanced sample could significantly vary the sensitivity for each group.

Thank you of allowing us to elucidate this further. We agree that the sensitivities could be influenced by the numerical disparity between the different cohorts. However, as mentioned previously, the number of images retrived was not based on race per se. We have acknowledged this in limitations.

“Unequal distribution in the different cohorts may have affected the performance of the AI model for each race.”

Page 10, Discussion

“The major leap in this work from our previous reports is the improvement in the detection of ICRB

group A tumors from 56% to 74%. While other studies report an accuracy of 82% to 100% based on

various models, the data on the AI model's efficacy for tumors of different ICRB groups was limited.”

The discussion seems to focus more on group A, because the AI model has managed to increase its

sensitivity compared to the past in this group: it could be better to highlight it in the title and also

in the summary, to offer a more correct perspective about the study itself.

Thank you for this valuable input. As rightly pointed out by the reviewer, we did attempt to focus on detection of early tumors.

However, that was not the only cohort we analyzed. We have highlighted this in the limitations. We are unable to add this to the title as early tumors form a subset of the entire cohort.

“The second limitation of this study was that despite acquiring a large cohort of images, the numbers in ICRB Groups A and B were smaller. To employ this technology as a screening tool needs improvement and further training on these sub-cohorts.”

Page 10, Discussion

“The color of a target object and its background can significantly affect the accuracy of AI models, and this has been shown with the detection of objects such as medicine pills [21], and in real-time fundus images [22]. Thus, the background fundus color is a crucial factor that needs to be considered while employing AI-based tools for early detection of RB in a community setting.”

This part of the text highlights once again how the color of the fundus based on ethnicity is a crucial factor in using the artificial intelligence. therefore it results in discrediting the study that has a too significant ethnic disparity

We agree with the author’s comment and we bring this out in the paper in: 1) Methods- Identification of specific problems and development of solutions and 2) Table 4.

We have elaborated in limitations that unequal numbers within different races is a possible confounder and these under-represented cohorts need to be further studied.

Methods:

“The color segmentation and contrast enhancement approaches were used only to extract features such as blood vessels and hemorrhages. We retrained the models for other fea-tures to capture other variabilities, as a standard practice described in the literature [26].”

“Unequal distribution in the different cohorts may have affected the performance of the AI model for each race. Thus, this AI model‘s generalizability beyond Asian Indians and Whites is still limited. On the same lines, whether a screening program based on this model needs to be race-specific or universal cannot be answered yet. Multicentric data-sharing studies are war-ranted to bridge these lacunae, especially for underrepresented groups and groups not represented in this study.”

Page 4, Methods and

Page 10, Discussion

“Nevertheless, in both scenarios, the AI model’s output was affected by race, and to improve accuracy, the AI models need to be trained from diverse cohorts.”

In this sentence it is specified the importance to have multiracial patients because the ethnicity is fundamental for the AI models.

Thank you for acknowledging this.

None

--

“This is a first-of-its-kind study to explore the ability of an AI model to detect and classify RB in a multiracial cohort. Because the study was based on data from a single Ocular Oncology Center, the distribution of patients between different racial cohorts was not uniform, which was a drawback.”

The conclusions present the limitations of the study: namely a non-uniform distribution of patients among the various races, with a preponderance of the white race, which is a little bit in contrast with the rest of the article. In fact, in it, little was said about the problems related to the data used for the following research.

Thank you for this suggestion. We have added these limitations to the discussion.

We have moved this sentence to limitations and elaborated further. “Because the study was based on data from a single Ocular Oncology Center in the United States, most of the patients were White [23]. Unequal distribution in the differ-ent cohorts may have affected the performance of the AI model for each race. Thus, this AI model‘s generalizability beyond Asian Indians and Whites is still limited. On the same lines, whether a screening program based on this model needs to be race-specific or universal cannot be answered yet. Multicentric data-sharing studies are warranted to bridge these lacunae, especially for underrepresented groups and groups not repre-sented in this study.”

Page 10, Discussion

To better enhance the study, it is advisable to review its form, in order to make the content more coherent. Since the cohort considered is unequal, it could be better to specify from the beginning the limits of the study due to numeric differences between races, so as to show the interesting work carried out without having to evidence the contradiction only at the end. Furthermore, since the cohort is mostly made up of white patients, the study could be focused differently: after a brief introduction explaining the cohort examined, the paper could concentrate on this created subgroup, analyzing the main aspects that emerged.

Thank you for this recommendation. We have clarified in the Methods section.

“The information about the patient’s race was retrieved from the medical records as re-ported by the parent or caregiver. Per institute protocols at the Ocular Oncology Ser-vice, Wills Eye Hospital, Philadelphia, U.S.A, race was recorded as White, Hispanic, African American, Asian Indian, Asian Oriental, or Others.

As of March 2024, the Statistical Policy Directive no. 15 on standards for maintain-ing, collecting, and presenting federal data on race and ethnicity combined race and ethnicity into one single question and listed the following categories: American Indian or Alaska Native, Asian, Black or African American, Hispanic or Latino, Middle Eastern or North African, Native Hawaiian or Pacific Islander, and White [22, 23]. We largely adhered to this format. Although “Hispanic” is considered more appropriate as an eth-nonym than a race, this cohort was deemed distinct to the study since their fundus pigmentation is known to vary from the White population [22-25]. We included consec-utive patients of all races, and heterogeneity was expected in the number of patients of different races, given the predominance of White patients seeking services at Wills Eye Hospital, Philadelphia, U.S.A.”

Page 2, Methods

Substantially this study represents the first step towards an increasingly widespread application of AI model for RB. A further improvement in the accuracy and specificity of that model would ensure greater safety and speed in its diagnosis, including in the most difficult cases, thanks to a more equitable multiracial cohort. In general, The AI model is intended to be used increasingly in the medical field, because it offers a valid help in scientific research.

Thank you for acknowledging this.

None

--

Reviewer 4 Report

Comments and Suggestions for Authors

Thanks the authors for writing up this AI related ophthalmology research. The idea is excellent, and the results are promising.

The title is a bit misleading, as there is no Paper II in the literature searchable by the peer reviewers.I would suggest to change "Artificial Intelligence and Machine Learning in OcularOncology, Retinoblastoma (ArMOR), Paper III: Experience with a Multiracial Cohort" to "Artificial Intelligence and Machine Learning in OcularOncology, Retinoblastoma (ArMOR): Experience with a Multiracial Cohort"

Most references cited in this manuscript appropriate and relevant to this research, excpet 1. There is redundancy for citing references

12. Kaliki, S.; Vempuluru, V.S.; Patil, G.; Viriyala, R.; Dhara, K.K. Artificial intelligence and machine learning in ocular oncology,
retinoblastoma (ArMOR): Paper II [Unpublished data]

Firstly, it should not be cited, as it is not published yet. The readers and reviewers cannot access any part of this manuscript, and thus could not serve the function of citation. Basically, it is a self citation only.

Secondly, there is no solid data cited from this unpublished manuscript. Looking into the manuscript text where reference [12] were cited, they are all general comments. Omitting [12] would affect the validity of the paper. Reference [11] is already enough for the citation to backup the percentage claim in the statement.

If the authors think that it is really needed for citation, please follow the author instructions.

According to Cancers journal's author instructions:

Free Format Submission

Cancers now accepts free format submission:

  • Your references may be in any style, provided that you use the consistent formatting throughout.
  • Self-citation (authors cite their own work) should be controlled within 10%. Justifications are required if there is more than 10%.
  • Unpublished materials intended for publication:
    4. Author 1, A.B.; Author 2, C. Title of Unpublished Work (optional). Correspondence Affiliation, City, State, Country. year, status (manuscript in preparation; to be submitted).
    5. Author 1, A.B.; Author 2, C. Title of Unpublished Work. Abbreviated Journal Name year, phrase indicating stage of publication (submitted; accepted; in press).
  • Unpublished materials not intended for publication:
    6. Author 1, A.B. (Affiliation, City, State, Country); Author 2, C. (Affiliation, City, State, Country). Phase describing the material, year. (phase: Personal communication; Private communication; Unpublished work; etc.)

For this manuscript, the references are not consistent in format in the manuscript, e.g.

reference 5. Strijbis, V.I.J.; de Bloeme, C.M.; Jansen, R.W.; Kebiri, H.; Nguyen, H.G.; de Jong, MC.; Moll, A.C.; Bach-Cuadra, M.; de Graaf,
P.; Steenwijk, M.D. Multi-view convolutional neural networks for automated ocular structure and tumor segmentation in retinoblas-
toma. Sci Rep. 2021;11(1):14590.

is of different format to reference

6. Rahdar, A.; Ahmadi, M.J.; Naseripour, M.; Akhtari, A.; Sedaghat, A.; Hosseinabadi, V.Z.; Yarmohamadi, P.; Hajihasani, S.;
Mirshahi, R. Semi-supervised segmentation of retinoblastoma tumors in fundus images. Sci Rep.2013;13, 13010.

Whereas reference
7. Jebaseeli, T.; David D. Diagnosis of ophthalmic retinoblastoma tumors using 2.75 D CNN segmentation technique. In Computa-
tional Methods and Deep Learning for Ophthalmology; Academic Press: Cambridge, MA, USA, 2023; pp. 107–119.

is of different format to reference

14. Global Retinoblastoma Study group; Fabian, I.D.; Abdallah, E.; Abdullahi, S.U.; Abdulqader, R.A.; Adamou, B.S.; Ademola-
Popoola D.S.; Adio, A.; Afshar, A.R.; Aggarwal, P.; Aghaji, A.E.; Ahmad, A.; Akib, M.N.R.; Al Harby, L.; Al Ani, M.H.; Alakbarova,
A.; Portabella, S.A.; Al-Badri, S.A.F.; Alcasabas, A.P.A.; Al-Dahmash, S.A.; Alejos, A.; Alemany-Rubio, E.; Alfa Bio, A.I.; Alfonso
Carreras, Y.; Al-Haddad, C.; Al-Hussaini, H.H.Y.; Ali, A.M.; Alia, D.B.; Al-Jadiry, M.F.; Al-Jumaily, U.; Alkatan, H.M.; All-Eriks-
son, C.; Al-Mafrachi, A.A.R.M.; Almeida, A.A.; Alsawidi, K.M.; Al-Shaheen, A.A.S.M.; Al-Shammary, E.H.; Amiruddin, P.O.; An-
tonino, R.; Astbury, N.J.; Atalay, H.T.; Atchaneeyasakul, L.O.; Atsiaya, R.; Attaseth, T.; Aung, T.H.; Ayala, S.; Baizakova, B.; Bala-
guer, J.; Balayeva, R .;Balwierz, W.; Barranco, H.; Bascaran, C.; Beck Popovic, M.; Benavides, R.; Benmiloud, S.; Bennani Gue-
bessi, N.; Berete, R.C.; Berry, J.L.; Bhaduri, A.; Bhat, S.; Biddulph, S.J.; Biewald, E.M.; Bobrova, N.; Boehme, M.; Boldt, H.C.;
Bonanomi, M.T.B.C.; Bornfeld, N.; Bouda, G.C.; Bouguila, H.; Boumedane, A.; Brennan, R.C.; Brichard, B.G.; Buaboonnam, J.;
Calderón-Sotelo, P.; Calle Jara, D.A.; Camuglia, J.E.; Cano, M.R.; Capra, M.; Cassoux, N.; Castela, G.; Castillo, L.; Català-Mora,
J.; Chantada, G.L.; Chaudhry, S.; Chaugule, S.S.; Chauhan, A.; Chawla, B.; Chernodrinska, V.S.; Chiwanga, F.S.; Chuluunbat, T.;
Cieslik, K.; Cockcroft, R.L.; Comsa, C.; Correa, Z.M.; Correa Llano. M.G.; Corson, T.W.; Cowan-Lyn, K.E.; Csóka, M.; Cui, X.;
Da Gama, I.V.; Dangboon, W.; Das, A.; Das, S.; Davanzo, J.M.; Davidson, A.; De Potter, P.; Delgado, K.Q.; Demirci, H.; Desjardins,
L.; Diaz Coronado, R.Y.; Dimaras, H.; Dodgshun, A.J.; Donaldson, C.; Donato Macedo, C.R.; Dragomir M.D.; Du, Y.; Du Bruyn,
M.; Edison, K.S.; Eka Sutyawan, I.W.; El Kettani, A.; Elbahi, A.M.; Elder, J.E.; Elgalaly, D.; Elhaddad, A.M.; Elhassan, M.M.A.;
Elzembely, M.M.; Essuman, V.A.; Evina, T.G.A.; Fadoo, Z.; Fandiño, A.C., Faranoush, M.; Fasina, O.; Fernández, D.D.P.G.; Fer-
nández-Teijeiro, A.; Foster, A.; Frenkel, S.; Fu, L.D.; Fuentes-Alabi, S.L.; Gallie, B.L.; Gandiwa, M.; Garcia, J.L.; García Aldana,
D.; Gassant, P.Y.; Geel, J.A.; Ghassemi, F.; Girón, AV.; Gizachew, Z.; Goenz, M.A.; Gold, AS.; Goldberg-Lavid, M.; Gole, GA.;
Gomel, N.; Gonzalez, E.; Gonzalez Perez, G.; González-Rodríguez, L.; Garcia Pacheco, H.N.; Graells, J.; Green, L.; Gregersen,
P.A.; Grigorovski, N.D.A.K.; Guedenon, K.M.; Gunasekera, D.S.; Gündüz, AK.; Gupta, H.; Gupta, S.; Hadjistilianou, T.; Hamel,
P.; Hamid, S.A.; Hamzah, N.; Hansen, E.D.; Harbour, J.W.; Hartnett, M.E.; Hasanreisoglu, M.; Hassan, S.; Hassan, S.; Hederova,
S.; Hernandez, J.; Hernandez, L.M.C.; Hessissen, L.; Hordofa, D.F.; Huang L.C.; Hubbard, G.B.; Hummlen, M.; Husakova, K.;
Hussein Al-Janabi, A.N.; Ida, R.; Ilic, V.R.; Jairaj, V.; Jeeva, I.; Jenkinson, H.; Ji, X.; Jo, D.H.; Johnson, K.P.; Johnson, W.J.; Jones,
M.M.; Kabesha, T.B.A.; Kabore, R.L.; Kaliki, S.; Kalinaki, A.; Kantar, M.; Kao, LY.; Kardava, T.; Kebudi, R.; Kepak, T.; Keren-
Froim, N.; Khan, Z.J.; Khaqan, HA.; Khauv, P.; Kheir, W.J.; Khetan, V.; Khodabande, A.; Khotenashvili, Z.; Kim, J.W.; Kim, J.H.;
Kiratli, H.; Kivelä, T.T.; Klett, A.; Komba Palet, J.E.K.; Krivaitiene, D.; Kruger, M.; Kulvichit, K.; Kuntorini, M.W.; Kyara, A.;
Lachmann, E.S.; Lam, C.P.S.; Lam, G.C.; Larson, S.A.; Latinovic, S.; Laurenti, K.D.; Le, B.H.A.; Lecuona, K.; Leverant, A.A.; Li, C.; Limbu, B.; Long, Q.B.; López, J.P.; Lukamba, R.M.; Lumbroso, L.; Luna-Fineman, S.; Lutfi, D.; Lysytsia, L.; Magrath, G.N.;
Mahajan, A.; Majeed, A.R.; Maka, E.; Makan, M.; Makimbetov, EK.; Manda, C.; Martín Begue, N.; Mason, L.; Mason, JO, 3rd.;
Matende, I.O.; Materin, M.; Mattosinho, C.C.D.S.; Matua, M.; Mayet, I.; Mbumba, F.B.; McKenzie, J.D.; Medina-Sanson, A.;
Mehrvar A.; Mengesha, A.A.; Menon, V.; Mercado, G.J.V.D.; Mets, M.B.; Midena, E.; Mishra, D.K.C.; Mndeme, F.G.; Mohame-
dani, A.A.; Mohammad, M.T.; Moll, A.C.; Montero, M.M.; Morales, R.A.; Moreira, C.; Mruthyunjaya, P.; Msina, M,S.; Msukwa,
G.; Mudaliar, S.S.; Muma, K.I.; Munier, F.L.; Murgoi, G.; Murray, T.G.; Musa, K.O.; Mushtaq, A.; Mustak, H.; Muyen, O.M.;
Naidu, G.; Nair, AG.; Naumenko, L.; Ndoye Roth, PA.; Nency, Y.M.; Neroev, V.; Ngo, H.; Nieves, R.M.; Nikitovic, M.; Nkanga,
E.D.; Nkumbe, H.; Nuruddin, M.; Nyaywa, M.; Obono-Obiang, G.; Oguego, N.C.; Olechowski, A.; Oliver, S.C.N.; Osei-Bonsu, P.;
Ossandon, D.; Paez-Escamilla, M.A.; Pagarra, H.; Painter, S.L.; Paintsil, V.; Paiva L.; Pal, B,P.; Palanivelu, M.S.; Papyan, R.; Par-
rozzani, R.; Parulekar, M.; Pascual Morales, C.R.; Paton, K.E.; Pawinska-Wasikowska, K.; Pe'er, J.; Peña, A Peric, S.; Pham, C.T.M.;
Philbert, R.; Plager, D.A.; Pochop, P.; Polania, R.A.; Polyakov, V.G.; Pompe, M.T.; Pons, J.J.; Prat, D.; Prom, V.; Purwanto, I.;
Qadir, A.O.; Qayyum, S.; Qian, J.; Rahman, A.; Rahman, S.; Rahmat, J.; Rajkarnikar, P.; Ramanjulu, R.; Ramasubramanian, A.;
Ramirez-Ortiz, M.A.; Raobela, L.; Rashid, R.; Reddy, M.A.; Reich, E.; Renner, L.A.; Reynders, D.; Ribadu, D.; Riheia, M.M.;
Ritter-Sovinz, P.; Rojanaporn, D.; Romero, L.; Roy, S.R.; Saab, R.H.; Saakyan, S.; Sabhan, A.H.; Sagoo, M.S.; Said, A.M.A.; Saiju,
R.; Salas, B.; San Román Pacheco, S.; Sánchez, G.L.; Sayalith, P.; Scanlan, T.A.; Schefler A.C.; Schoeman, J.; Sedaghat, A.; Sere-
gard, S.; Seth, R.; Shah, A.S.; Shakoor, S.A.; Sharma, M.K.; Sherief, S.T.; Shetye, N.G.; Shields, C.L.; Siddiqui, S.N.; Sidi Cheikh,
S.; Silva, S.; Singh, A.D.; Singh, N.; Singh, U.; Singha, P.; Sitorus, R.S.; Skalet, A.H.; Soebagjo, H.D.; Sorochynska, T.; Ssali, G.;
Stacey, A.W.; Staffieri, S.E., Stahl, E.D.; Stathopoulos, C.; Stirn Kranjc, B.; Stones, D.K., Strahlendorf, C.; Suarez, M.E.C.; Sultana,
S.; Sun, X.; Sundy, M.; Superstein, R.; Supriyadi, E.; Surukrattanaskul, S.; Suzuki, S.; Svojgr, K.; Sylla, F.; Tamamyan, G.; Tan, D.;
Tandili, A.; Tarrillo Leiva, F.F.; Tashvighi, M.; Tateshi, B.; Tehuteru, E.S.; Teixeira, L.F.; The, K.H.; Theophile, T.; Toledano, H.;
Trang, D.L.; Traoré F.; Trichaiyaporn, S.; Tuncer, S.; Tyau-Tyau, H.; Umar, A.B.; Unal, E.; Uner, O.E.; Urbak, S.F.; Ushakova, T.L.;
Usmanov, R.H.; Valeina, S.; van Hoefen Wijsard, M.; Varadisai, A.; Vasquez , L.; Vaughan, L.O.; Veleva-Krasteva, NV.; Verma, N.;
Victor, A.A.; Viksnins, M.; Villacís Chafla, E.G.; Vishnevskia-Dai, V.; Vora, T.; Wachtel, A.E.; Wackernagel, W.; Waddell, K., Wade,
P.D.; Wali, A.H.; Wang, Y.Z.; Weiss, A.; Wilson, M.W.; Wime.; A.D.C.; Wiwatwongwana, A.; Wiwatwongwana, D.; Wolley Dod,
C.; Wongwai, P.; Xiang, D.; Xiao, Y.; Yam, J.C.; Yang, H.; Yanga, J.M.; Yaqub, M.A.; Yarovaya, V.A.; Yarovoy, A.A.; Ye, H.; Yousef,
Y.A.; Yuliawati, P.; Zapata López, A.M.; Zein, E.; Zhang, C.; Zhang, Y.; Zhao, J.; Zheng, X.; Zhilyaeva, K.; Zia, N.; Ziko, O,A.O.;
Zondervan, M.; Bowman R. Global Retinoblastoma Presentation and Analysis by National Income Level. JAMA Oncol.
2020;6(5):685-695.

Please remove few references to keep in line with the self citation rate requirement.

Author Response

Reviewer 4

Thanks the authors for writing up this AI related ophthalmology research. The idea is excellent, and the results are promising.

Response: Thank you for the comments 

Comment

Response

Changes

Location

1. The title is a bit misleading, as there is no Paper II in the literature searchable by the peer reviewers.I would suggest to change "Artificial Intelligence and Machine Learning in OcularOncology, Retinoblastoma (ArMOR), Paper III: Experience with a Multiracial Cohort" to "Artificial Intelligence and Machine Learning in OcularOncology, Retinoblastoma (ArMOR): Experience with a Multiracial Cohort"

Thank you for this input. We have deleted “Paper 3”from the title.

Title: “Artificial Intelligence and Machine Learning in Ocular

Oncology, Retinoblastoma (ArMOR): Experience with a Multi-racial Cohort”

Page 1, Title

Most references cited in this manuscript appropriate and relevant to this research, excpet 1. There is redundancy for citing references

12. Kaliki, S.; Vempuluru, V.S.; Patil, G.; Viriyala, R.; Dhara, K.K. Artificial intelligence and machine learning in ocular oncology,

retinoblastoma (ArMOR): Paper II [Unpublished data]

Firstly, it should not be cited, as it is not published yet. The readers and reviewers cannot access any part of this manuscript, and thus could not serve the function of citation. Basically, it is a self citation only.

Secondly, there is no solid data cited from this unpublished manuscript. Looking into the manuscript text where reference [12] were cited, they are all general comments. Omitting [12] would affect the validity of the paper. Reference [11] is already enough for the citation to backup the percentage claim in the statement.

Thank you for this suggestion We have deleted this reference.

We have deleted this reference.

References, Page 11

Round 2

Reviewer 1 Report

Comments and Suggestions for Authors

The authors have made all the necessary adjustments. I believe the article is now ready for publication.

Comments on the Quality of English Language

None

Author Response

Comment: The authors have made all the necessary adjustments. I believe the article is now ready for publication.

Response: Thank you 

Reviewer 2 Report

Comments and Suggestions for Authors

I still have some major concerns about the aforementioned points.

Author Response

Comment 1: The author claim that the workflow used by the authors has already been described in their previous articles but as far as I can see for example reference 12 referes to unpublished data so I believe that the title should not be "III" since the second paper has not been published yet (Kaliki, S.; Vempuluru, V.S.; Patil, G.; Viriyala, R.; Dhara, K.K. Artificial intelligence and machine learning in ocular oncology, retinoblastoma (ArMOR): Paper II [Unpublished data]).

Response: This has already been addressed in our first revision 

Comment 2: I have some doubts about the classification used to identify White, African-American, Asian, and Hispanic patients: is there a specific scientific reference to frame patients according to this method? For example what about the label of “white” patient: could Caucasian be used as alternative? Again regarding “Hispanic” even though I believe it is easily possible from a cultural point of view to use this label I am not quite convinced that this could be used in a scientific article: which countries would be included under this definition? Please provide more robust scientific evidence about the choice to label patients under a certain racial category and the methodology followed to do so.

Response: Thank you for giving us the opportunity to clarify this again. The information of race was parent-reported. However, we adhered to the terminology prescribed by the United States Census Bureau in adherence to the 1997 Office of Management and Budget. Althouth “Hispanic”is more appropriately an ethnonym, we distinguished this cohort from the Caucasians due to differences in the fundus pigmentation. Further, the U.S. Office of Management and Budget's Statistical Policy Directive No. 15 lists Hispanic as a separate race.

We have added further information to further clarify the methodology employred for race. The cited reference for terminology on race ethnicity is attached for the reviewer’s perusal (Page 22191).

“Hispanics included individuals of Mexican, Puerto Rican, Salvadoran, Cuban, Domini-can, Guatemalan, and other Central or South American or Spanish culture or origin [22].

We have now clarified this in the abstract as well: “Of 210 eyes, 153 (73%) belonged to White, 37 (18%) to African-American, 9 (4%) to Asian, 6 (3%) to Hispanic races, based on the U.S. Office of Management and Budget's Statistical Policy Directive No.15 and 5 (2%) had no reported race.”

Reviewer 3 Report

Comments and Suggestions for Authors

My congratulations for the authors for their interesting work, carried out with accuracy and precision.

Author Response

Comment: My congratulations for the authors for their interesting work, carried out with accuracy and precision.

Response: Thank you for appreciating our work and efforts.